# The Impact of Muscarinic Antagonism on Psychosis-Relevant Behaviors and Striatal [^11^C] Raclopride Binding in Tau Mouse Models of Alzheimer’s Disease

**DOI:** 10.3390/biomedicines11082091

**Published:** 2023-07-25

**Authors:** Heidy Jimenez, Joseph Carrion, Leslie Adrien, Adam Wolin, John Eun, Ezra Cinamon, Eric H. Chang, Peter Davies, An Vo, Jeremy Koppel

**Affiliations:** 1The Litwin-Zucker Research Center for the Study of Alzheimer’s Disease, The Feinstein Institutes for Medical Research, Northwell Health, Manhasset, NY 11030, USA; hjimenez1@northwell.edu (H.J.); jcarrion1@northwell.edu (J.C.); ladrien@northwell.edu (L.A.); awolin26@fastmail.com (A.W.); johndavideun@gmail.com (J.E.); echang1@northwell.edu (E.H.C.); pdavies@northwell.edu (P.D.); avo@northwell.edu (A.V.); 2Department of Biochemistry, Queens College, Flushing, NY 11355, USA; cinamonezra@gmail.com

**Keywords:** tau, psychosis, scopolamine, locomotion, Alzheimer’s disease, hopping

## Abstract

Psychosis that occurs over the course of Alzheimer’s disease (AD) is associated with increased caregiver burden and a more rapid cognitive and functional decline. To find new treatment targets, studies modeling psychotic conditions traditionally employ agents known to induce psychosis, utilizing outcomes with cross-species relevance, such as locomotive activity and sensorimotor gating, in rodents. In AD, increased burdens of tau pathology (a diagnostic hallmark of the disease) and treatment with anticholinergic medications have, separately, been reported to increase the risk of psychosis. Recent evidence suggests that muscarinic antagonists may increase extracellular tau. Preclinical studies in AD models have not previously utilized muscarinic cholinergic antagonists as psychotomimetic agents. In this report, we utilize a human–mutant–tau model (P301L/COMTKO) and an over-expressed non-mutant human tau model (htau) in order to compare the impact of antimuscarinic (scopolamine 10 mg/kg/day) treatment with dopaminergic (reboxetine 20 mg/kg/day) treatment, for 7 days, on locomotion and sensorimotor gating. Scopolamine increased spontaneous locomotion, while reboxetine reduced it; neither treatment impacted sensorimotor gating. In the P301L/COMTKO, scopolamine treatment was associated with decreased muscarinic M4 receptor expression, as quantified with RNA-seq, as well as increased dopamine receptor D2 signaling, as estimated with Micro-PET [^11^C] raclopride binding. Scopolamine also increased soluble tau in the striatum, an effect that partially mediated the observed increases in locomotion. Studies of muscarinic agonists in preclinical tau models are warranted to determine the impact of treatment—on both tau and behavior—that may have relevance to AD and other tauopathies.

## 1. Introduction

Psychosis, which affects nearly 40% of those with Alzheimer’s disease (AD) over the course of their illness [1], is associated with an increased burden to caregivers, who are often the unfortunate target of violence and aggression [2,3]. It also heralds a more rapid decline, higher rates of skilled nursing placement, and a hastened mortality [4]. The results of investigations carried out by our lab and others, utilizing cerebrospinal fluid, [5], postmortem neuropathology [6,7,8], and [^18^F]-flortaucipir (AV1451) positron emission tomography (PET) [9], suggest that the severity and distribution of tau pathology is a mediator of both psychosis and decline. There are no medications with FDA approval for the treatment of this condition, and available antipsychotics developed for the treatment of schizophrenia, which are frequently employed in AD psychosis [4], have a black box warning from the FDA related to an increased risk of cardiovascular events and mortality [10,11]. While new treatments focused on novel targets are desperately needed, preclinical models with which to test them are lacking.

An obstacle to drug development in AD psychosis has been the lack of appropriate preclinical models to test novel therapies [12]. Drug development pipelines for clinical trials in AD utilize transgenic neuropathologic animal models, targeting cognitive phenotypes to test their compounds, before going into humans [13], while antipsychotic agents are developed by targeting relevant behavioral phenotypes in the models: locomotor hyperactivity (often drug-induced) and sensorimotor gating [12,14,15,16]. Sensorimotor gating is an involuntary faculty that reflects the capacity to attend to an ongoing sensory stimulus and gate the motor response to a subsequent distracting startle stimulus; prepulse inhibition (PPI) of the acoustic startle reflex—the dampening of an automatic startle-induced contracture of skeletal muscle that represents a defensive posture [17] with an ongoing (prepulse) stimulus—is an experimental model used for the quantification of sensorimotor gating [18]. While PPI has many experimental applications not limited to psychosis [19], it is particularly relevant in psychotic illnesses in which the dopamine system is disrupted, such as schizophrenia, bipolar disorder, and intoxication-induced psychosis [20,21,22], but it is impaired in dementia syndromes, especially those with psychotic phenotypes [23,24]. In rodents, compounds that make people psychotic, such as anticholinergics [25,26] and dopamine agonists, acting at dopamine D2 receptors (D2R) [27], disrupt PPI [28,29,30,31]. 

Abnormalities of dopamine and cholinergic neurotransmission have been implicated in AD psychosis. Alterations in striatal dopamine D3 receptor (D3R) expression have been observed in psychotic AD [32], while the occupancy of the striatal D2R with antipsychotic alterations determines a therapeutic window of treatment response [33]. While substance-induced surges in extracellular dopamine precipitate psychosis in the context of acute and chronic exposure to methylphenidate [34], cocaine [35], and the iatrogenic psychosis of Parkinson’s disease following treatment with levodopa [36], surprisingly, the use of dopaminergic psychostimulants in AD for the treatment of behavioral disorders have not been associated with an increased risk of psychotic symptoms [37,38]. Psychosis in AD is also believed to be related to abnormalities in cholinergic neurotransmission [39], and elevations in the muscarinic receptor binding have been reported in the cortices of AD patients with psychosis; [40] unlike dopaminergic therapies, psychosis in AD has been associated with the use of anticholinergic medications [41], with exposure more than doubling the risk of incident psychosis [42]. Cholinesterase inhibitors utilized in the cognitive treatment of AD have been shown to decrease the emergence of psychotic symptoms [43,44,45]. In the limited literature that exists on the use of pharmacology to model psychosis in AD, a psychotomimetic challenge in preclinical models has been accomplished with dopamine-promoting stimulants [12,15,46], rather than compounds that target the cholinergic system, as epidemiologic evidence would recommend. The antimuscarinic scopolamine has been used in AD, as an induction agent, in memory paradigms only [47], mimicking AD-like impairments in preclinical [48] and acute neurocognitive performance models of AD [49], but it has not previously been employed as an induction model of AD psychosis-relevant behaviors.

As tau pathology in the brain has been demonstrated to have transdiagnostic relevance to psychosis, including in AD, [9], psychosis secondary to traumatic brain injury [50], and late-life psychotic depression [51], we have investigated the ability of tau to disrupt behaviors in mice, with relevance to human psychosis, that can be modeled in rodents—spontaneous locomotion, startle response, and sensorimotor gating [12]. While the tauopathy of AD has been well characterized, as there are no known autosomal dominant mutations in the microtubule-associated protein tau (*MAPT*) gene that are associated with familial AD, transgenic models that express mutant forms of human tau, associated with frontotemporal dementia [52], have been employed to study outcomes with relevance to the disease [53]. Our previous studies suggest that total tau and pathogenic phosphotau species disrupt PPI in the rTg(P301L)4510 model of tauopathy [54], and increases in tau expression may be associated with changes in locomotion [55]. We have developed a preclinical tau model of psychotic AD, expressing the P301L human mutant tau gene associated with human tauopathy [56], together with the deletion of the catechol-o-methyltransferase gene (COMT), in order to drive dopamine neurotransmission [57]. The P301L/COMTKO model manifests increased extracellular dopamine [58], and it evidences impaired PPI and hyperlocomotion relative to wildtype littermate controls [55]. 

In the current report, in order to determine whether muscarinic antagonism has relevance as a psychotomimemtic agent in preclinical models of AD tauopathy, we investigate the effects of treatment in two tau models: the P301L/COMTKO [58] and the htau [59] model. In tests of spontaneous locomotion and sensorimotor gating in these models, we compare the antimuscarinic scopolamine with reboxetine—a norepinephrine reuptake inhibitor that we have previously proven to increase extracellular dopamine in the frontal cortex of P301L mice, especially those with a COMT deletion [58]. As both dopaminergic [58] and, more recently, muscarinic receptors [60] have been implicated in tau neurobiology, we followed the behavioral experiments with an investigation into the impact of scopolamine on dopaminergic and muscarinic pathway genes, as well as tau biochemistry, in the P301L/COMTKO model. Guided by the results of these experiments, in order to evaluate the impact of cholinergic antagonism on dopaminergic signaling in the striatum—a hub for locomotion and sensorimotor gating—we utilized [^11^C] raclopride Micro-Positron Emission Tomography (PET) in P301L/COMTKO mice to estimate D2/D3 receptor occupancy and, thus, endogenous dopamine levels following scopolamine treatment.

## 2. Materials and Methods

### 2.1. Animals

All experiments were approved by the Institutional Animal Care and Use Committee (IACUC) at the Feinstein Institutes for Medical Research (FIMR). Mice were kept on a reverse light/dark 12 h cycle, and testing was performed in a sound-attenuated room. In order to evaluate the effects of dopaminergic and cholinergic neuropharmacology on behavior and neurophysiology, we utilized P301L/COMTKO [58] mice developed in our lab, which express a human mutation (P301L) in the microtubule-associated protein tau (*MAPT*) gene that causes frontotemporal dementia and Parkinsonism linked to chromosome 17 (FTDP-17), resulting in motor and behavioral deficits in mice driven by neurofibrillary tangle pathology [56] in the context of the deletion of the COMT gene [55]. P301L/COMTKO mice were bred and maintained at the Center for Comparative Physiology in the FIMR. In addition, for behavioral experiments, we also utilized htau mice, which represent a less severe model of over-expressed non-mutant human tau (with a competent COMT gene) that develops behavioral deficits somewhat later than P301L models [59]. Htau mice were obtained from The Jackson Laboratory Tg(*MAPT*)8c**Pdav**/J, (strain #005491, Bar Harbor, ME, USA) and housed at the FIMR for 1 month prior to testing. There were 45 female 4-month-old P301L/COMTKO and 45 female 6-month-old htau mice utilized in these studies. Tau pathology accrues at different rates and begins at different ages in P301L-based and htau models (htau later), and the behavioral experiments were designed to take place after pathology begins to develop but significantly prior to the onset of any motor impairments from posterior and spinal tau pathology that could impact locomotor assessments [59,61]. We utilized female mice for comparisons, as tau transgenic expression is sex dependent, with females exhibiting more aggressive pathology [62], and as drug-induced changes in locomotion and sensorimotor gating have been shown to be sex-dependent [14]. 

### 2.2. Pharmacology

In order to assess changes in behavior, with relevance to psychosis, and correlate these with changes in gene expression and tau pathology, cohorts of 15 female P301L/COMTKO mice and 15 female htau mice were randomized and treated under each of the following three conditions: reboxetine (Thermo Fisher Scientific, Waltham, MA, USA) 20 mg/kg/day IP for 7 days; scopolamine (Sigma-Aldrich, St. Luis, MO, USA) 10 mg/kg/day for 7 days; saline vehicle IP, once daily, for 7 days. Treatments of 7 days were employed in order to allow changes in gene expression and tau phosphorylation that could impact behavior, as well as to model the psychosis that emerges in people in response to medication exposure that happens over several days of treatment rather than from a single dose [41,42]. As we have shown that reboxetine induces surges of cortical dopamine, the doses were selected at 20 mg/kg in P301L/COMTKO mice [58], while scopolamine—a muscarinic antagonist—has been shown to reliably induce PPI deficits in wildtype mice at 10 mg/kg [26]. On the seventh day, at 90–150 min post-injections of pharmacology, we conducted behavioral experiments or MicroPET, followed by sacrifice, for tau biochemistry/RNA-Seq.

### 2.3. Locomotor Activity/Open Field

In order to compare the impact of dopaminergic stimulation and cholinergic antagonism on spontaneous locomotion in free-moving mice, mice were tested in an open acrylic box (17″ × 17″) with 12″ walls, allowing them to freely explore the environment for 10 min while being recorded by video tracking software. As we have previously published [55], mice underwent 10 min open field testing utilizing video tracking, with EthoVision XT Mouse Behavioral Recognition tracking software, via a Gigabit Ethernet (Gig-E) high resolution camera (Noldus Information Technology, Lessburg, VA, USA) that quantifies velocity (distance traveled/time) and other behavioral syllables, including hopping. Immediately following the baseline assessment, mice were treated under the three treatment conditions: reboxetine for 20 mg/kg/day, scopolamine for 10 mg/kg/day, or saline for 1 week. After 1 week of treatment, 60 min following the last injection, mice were assayed, again, with open field testing in order to calculate percent changes in locomotive velocity for each animal (%Δ from mean baseline measured in inches/second), as in previously published methodology [55]. In addition to locomotor velocity, whose definition is the distance traveled over a fixed period of time (10 min in open field), EthoVision XT quantifies the frequency of hopping—a gait that is characterized by bilateral and symmetrical forelimb/hindlimb synchronous movement [63]—which is qualitatively different from walking and reflects a brief burst of activity [64]. We utilized this metric as an assessment of qualitative changes in gait that are separate from distance traveled that may be the result of alteration in central locomotor networks related to pharmacology. 

### 2.4. Acoustic Startle and Sensory Motor Gating

For the interrogation of sensorimotor gating, at baseline and 90 min following the last injection, an SR-LAB system acoustic startle box with digitized electronic output, comprising a piezoelectric accelerometer mounted under a Plexiglass cylinder and integrated with startle response software (San Diego Instruments, San Diego, CA, USA), was utilized to generate startle and measure PPI in all treated mice. As in our previously reported methodology [54], PPI assessment began with each animal being acclimated to the startle box 2 weeks prior to the PPI sessions for 10-min acclimation sessions at a background noise intensity of 65 dB. For PPI sessions, the SR-LAB machine was programmed to deliver acoustic startle stimuli over a 65 dB background with a variable inter-trial interval. The startle stimulus was presented as a fast-rise noise burst lasting 40 milliseconds at an intensity of 120 dB. The animal’s whole-body flinch response to each stimulus was recorded as 65 consecutive 1 millisecond recordings at the stimulus onset. For prepulse trials, a prepulse of 12 dB over the background (77 dB), for a 20 millisecond duration, preceded the primary pulse by 100 milliseconds. Acoustic startle reactivity in millivolts was calculated from the average startle magnitude for the initial 6 pulses alone (120 dB) trials. The PPI—the percent inhibition of the acoustic startle amplitude when preceded by a prepulse—was calculated for each prepulse intensity (12 dB) as
(1)$$100\times \frac{pulse\;alone\;amplitude}{prepulse\;with\;pulse\;amplitude}$$
where scores were averaged across a pseudorandom admixture of trials, as in previously published methodology [54].

### 2.5. Bulk RNA-Seq

In order to evaluate differentially expressed genes associated with treatment, as in our previously published work [65], following sacrifice after 1 week of exposure, total RNA was extracted from the brains of scopolamine-treated P301L/COMTKO mice and saline-treated P301L/COMTKO mice with an RNeasy Mini Kit (Qiagen, Hilden, Germany), according to the manufacturer’s protocol [65]. NanoDrop ND-100 Spectrophotometer was used to determine the concentration of RNA samples (NanoDrop Technologies, Wilmington, DE, USA), and BioAnalyzer RNA 2100 kit was used to test their integrity (Agilent Technologies, Santa Clara, CA, USA) [65]. RNA sequencing was performed with the Illumina mRNA TruSeq Stranded method on Illumina HiSeq (Illumina, San Diego, CA, USA) [65]. We used the RNA-seq Alignment app and Illumina BaseSpace Sequence Hub, together with the DRAGEN (Dynamic Read Analysis for Genomics) platform (version 3.9.0), to calculate the differential expression of genes in P301L/COMTKO mice under the two different treatment conditions. Differential expression analysis was performed using DESeq2 to calculate log_2_-fold-change (LFC) estimates [66]. The UCSC mm10 *mus musculus* (with RefSeq gene annotation) was the reference genome. Wald test *p* values below 0.05 for log_2_-fold-change (LFC) estimates of scopolamine-versus-saline treated mice were tested for false discovery in multiple comparisons utilizing the Benjamini–Hochberg procedure [65].

### 2.6. MicroPET

#### 2.6.1. Image Acquisition 

MicroPET imaging with [^11^C] raclopride was performed to estimate striatal dopamine availability via D2R occupancy [67]. Upon arrival into the imaging suite, animals were acclimated for 1 h prior to anesthesia induction with a 2–2.5% isoflurane/oxygen mixture. After induction, animals were quickly positioned onto the Inveon^®^ MicroPET camera gantry using a customized two-animal nose cone (in head-to-head configuration) to maintain gaseous anesthesia. After confirmation of anesthesia, the tail vein of each rodent was cannulated using a customized 30 G catheter, which was flushed with 0.2 mL of heparinized physiological saline. After confirmation of each cannula patency, the line was lightly secured with micropore tape. Using manual controls, the laser cross hair was fixed on the center of the nose cone, and the platform was automatically positioned in the center of the field of vision (FOV), with a vertical height of 17 mm used for all acquisitions. After centering, the image acquisition workflow (IAW, Siemens) software was cued to await the delivery of the [^11^C] raclopride radiotracer, which was synthesized in the cyclotron radiochemistry facility adjacent to the MicroPET suite. The vial was transported and placed into a fully shielded dosing station. With the imaging suite, 2 doses of ^11^C-raclopride were drawn into a syringe, measured in a Capintec dose calibrator to 0.5~0.8 mCi, and transported, within a leaded syringe holder, to the MicroPET camera, where two people coordinated to administer the doses. At the signal, approximately 0.5~0.8 mCi (in 0.2 mL) was slowly injected via the tail vein with the simultaneous start of a 90 min dynamic imaging acquisition. Anesthesia was maintained at an approximate 1.5–2% isoflurane/oxygen mixture throughout the 90 min, as well as for the 10 min transmission scan that followed. After the completion of the [^11^C] raclopride scan (120 min following the injection of [^11^C] raclopride to allow for the elimination of the isotope), a dose of approximately ~0.5 mCi of [^18^F] FDG (in 0.3 mL) was injected intraperitoneally (i.p.), and 35–40 min allowed for uptake of the [^18^F] FDG tracer, followed by a 10 min static scan. All scans were acquired and converted into histograms using Inveon Acquisition workflow (IAW 1.5). The [^11^C] raclopride histograms were parceled into 49 frames, totaling 5400 s (90 min), in the following way: [^11^C] raclopride emission histogram bins: 10 bins × 60 s; 10 bins × 60 s; 5 bins × 120 s; 2 bins × 300 s; 2 bins × 300 s; 5 bins × 120 s; 5 bins × 120 s; 5 bins × 120 s; 5 bins × 120 s. The [^18^F] FDG scans were acquired and histogrammed into a single 600 s bin. Transmission scans were acquired immediately following emission acquisition with an internal [59] cobalt source, and they were used for attenuation. Reconstruction of both [^11^C] raclopride and [^18^F] FDG scans was done using an ordered subset expectation 3D maximum a posteriori (OSEM3D/SP-MAP), Ordered Subset Expectation Maximization (OSEM) of 18 iterations, and a target resolution of 1.5 mm. After reconstruction, raw images were bounding box aligned, skull stripped, and dose/weight corrected. 

#### 2.6.2. Image Analysis

To align [^11^C] raclopride brain images into anatomical space, [^18^F] FDG scans of the striatum (acquired without repositioning after [^11^C] raclopride) were used in the following way. Each unprocessed [^18^F] FDG image was first opened using the fusion toolbox within Pixel-Wise Modeling Software (PMOD 4.4) and was sorted top to bottom in the coronal plane to separate the individual images acquired in head-to-head configuration. The [^18^F] FDG scans were co-registered to an [18F] ^FDG^ template aligned in Paxinos and Franklin coordinates [68], and the individual 3-D transformations from the template-aligned [^18^F] FDG scans were saved. The dynamic [^11^C]-raclopride images were loaded in PMOD and sorted in top/bottom coronal configuration, with units set to Kilobecquerel/cubic centimeters (kBq/cc). Loading operations were configured for the averaged activity/mean by the number of frames [49]. Without manual manipulation, the [^18^F] FDG transformation for each rodent was applied to its corresponding [^11^C]-raclopride scan. This placed all [^11^C]-raclopride within a defined “bounding box” and allowed for further preprocessing, which included dose/weight correction and skull striping within PMOD 4.4.

To evaluate the impact of scopolamine on dopaminergic neurotransmission at the D2R in ventral and dorsal striatum, region-of-interest (ROI) analysis was performed for the [^11^C]-raclopride PET images of scopolamine-treated mice and saline-treated mice by using Waxholm Space (WHS) Atlas [69] and in-house MATLAB scripts. The [^11^C] raclopride PET images of these two groups were registered to a C57BL/6J mouse MRI template [68]. The WHS T1 was also registered to a C57BL/6J mouse MRI template, and then, the affine transformation matrix was applied to the WHS atlas to be in the same space as [^11^C] raclopride PET images. The mean values of [^11^C]-raclopride PET in caudate-putamen (WHS 23, Appendix A; [69] red) and nucleus accumbens (WHS 32, Appendix A [69] black) were measured and normalized by the mean value of cerebellum (WHS 35 [69]) in each image. 

Furthermore, we performed whole brain voxel-wise analysis using SPM-Mouse software (The Wellcome Centre for Human Neuroimaging, UCL Queen Square Institute of Neurology, London, UK, https://www.fil.ion.ucl.ac.uk/spm/ext/#SPMMouse, accessed on 28 March 2023) [70] to identify significant regions in anatomical space, in which [^11^C]-raclopride PET values decrease in scopolamine-treated mice, relative to saline-treated mice. Inter-subject variability in imaging data was accounted for by dividing each [^11^C]-raclopride PET scan by its mean value of cerebellum as a normalization factor. 

#### 2.6.3. Tau ELISA

We utilized a high-sensitivity EILISA technique, developed in our lab, to quantify the total soluble tau (μg/mg protein) by utilizing monoclonal capture and detection tau antibodies [71]. Following sacrifice, as in our previously published work, the brains of P301L/COMTKO mice were removed, the forebrain was dissected from the hindbrain, and the striatum was dissected and prepared for quantitative tau biochemistry [65,72]. The striatum was homogenized using an appropriate volume of homogenizing buffer comprised of a solution of Tris-buffered saline, pH 7.4, containing: 10 mM NaF, 1 mM NaVO3, and 2 mM EGTA, with a complete Mini protease inhibitor cocktail (Roche Molecular Biochemicals, Indianapolis, IN, USA). The prepared samples were stored at −80 °C. Heat stable preparations were used to obtain soluble tau levels by first adding 5% β-Mercaptoethanol and 200 mM NaCl to the brain homogenate. Samples were then heated at 100 °C for 10 min, cooled at 4 °C for 15 min on ice, and then centrifuged at 1300× *g* at 4 °C for 15 min, followed by supernatant collection. The 96 well plates were coated with DA31 tau capture antibody, at a concentration of 6 ng/mL in the coating buffer, for at least 48 h at 4 °C. Plates were washed 3× in the wash buffer, and they were blocked for 1 h at 20 °C using Starting Block in order to prevent non-specific binding. Each plate was then washed 5×, and 50 μL of the sample was added to the wells of the plate, with 50 μL of DA9 tau detection antibody. Plates were then incubated overnight at 4 °C and washed 9× in the wash buffer. The 1-Step ULTRA TMB-ELISA (Thermo Fisher, Waltham, MA, USA) was added, for 30 min at 20 °C, before stopping the reaction with 2 MH_2_SO_4_. Plates were read with a TECAN Infinite m200 (TECAN, Mannedorf, Switzerland) plate reader at 450 nm, and tau was quantified in a relationship to the total protein concentration (μg/mg).

### 2.7. Statistics

GraphPad Prism version 9.5.0 was utilized for statistical calculations. For group comparisons in locomotion and acoustic startle experiments, ordinary one-way ANOVA, followed by Tukey’s multiple comparisons test, was utilized with adjusted *p* < 0.05 to be considered significant. For PPI treatment effects in high-baseline inhibitors, after dividing the baseline measurements of both genotypes of tau mice at the median and selecting those mice at and above the median, repeated measures of two-way ANOVA were employed. Multiple linear regression was employed to evaluate the relationship between the pharmacology treatment group and continuous variables (tau and distance traveled in the open field). For statistical analysis in imaging experiments, all [^11^C]-raclopride images were group realigned to the FDG mouse brain template using the Statistical Parametric Mapping (SPM5) mouse toolbox within MATLAB. After realignment, images were smoothed with an isotropic Gaussian kernel FWHM (full width at half maximum)—0.56 mm in all directions—prior to the analysis. The differences in ^11^C-raclopride relative uptakes, between scopolamine mice and saline mice, were assessed using Student’s *t*-tests. Group differences were considered significant at a voxel-level threshold of *p* < 0.01. Individual data from the intersection of the significant regions and either caudate-putamen or nucleus accumbens were measured with post-hoc ROI analyses using in-house MAPLAB scripts. Group differences were considered significant at *p* < 0.05 (2-tailed).

## 3. Results

### 3.1. Behavior

#### 3.1.1. Locomotion

There were significant differences in change of velocity (% change in total distance traveled/time over baseline) between the treatment groups in both P301L/COMTKO (Figure 1A, ANOVA, F(2,41) = 56.08, *p* < 0.0001) and htau models (Figure 1B, ANOVA, F(2,42) = 17.21, *p* < 0.0001). Pairwise comparisons (Tukey’s multiple comparison test) revealed that, in P301L/COMTKO mice (Figure 1A, *p* < 0.0001) and htau mice (Figure 1B, *p* = 0.0008), scopolamine significantly increased velocity from the baseline in comparison with saline-treated mice. Reboxetine decreased velocity in comparison with saline-treated P301L/COMTKO (Figure 1A, *p* = 0.046) but not htau (Figure 1B) mice. Further, in P301L/COMTKO and htau mice, changes in the frequency of hopping with treatment were significantly different between treatment groups (Figure 1C, ANOVA, F(2,42) = 7.044, *p* = 0.0023), (Figure 1D, ANOVA F(2,42) = 11.81, *p* < 0.0001), respectively. Pairwise comparisons (Tukey’s) revealed that scopolamine increased hopping, in comparison to the saline treatment, in both tau models (Figure 1C, *p* = 0.0048) (Figure 1D, *p* = 0.0043). Reboxetine did not impact hopping, in comparison to the saline-treated mice, in either model. In order to drill down into the relationship between hopping and locomotor activity in response to scopolamine—to determine whether increases in hopping represent a change in gait or if they are simply artifacts of increased distance traveled with more opportunities for hopping at equivalent frequencies—we calculated hopping frequency (number)/distance traveled (inches) in each mouse over the course of the open field assessment and compared scopolamine-treated and saline-treated mice. Scopolamine increased hopping frequency/distance traveled in P301L/COMTKO mice and htau mice, which is a trend that approached but did not achieve significance in either model (Figure 1C *t*(28) = 1.58, *p* = 0.06) (Figure 1D *t*(28) = 1.40, *p* = 0.09), respectively, suggesting that scopolamine treatment may have effectuated a qualitative change in gait from walking and meandering to hopping—a more explosive movement pattern.

#### 3.1.2. Acoustic Startle/PPI

Scopolamine increased the acoustic startle amplitude from the baseline in both the P301L/COMTKO and htau mice, while reboxetine decreased the acoustic startle amplitude (Figure 2A,B). In the P301L/COMTKO model, there were significant treatment group differences (Figure 2A, ANOVA, F(2,42) = 8.228, *p* = 0.001). Pairwise comparisons revealed that scopolamine significantly increased the mean acoustic startle in comparison to reboxetine (Tukey’s, *p* = 0.0006). In the htau model, there were significant treatment group differences (Figure 2B, ANOVA, F(2,42) = 3.213, *p* = 0.05), but there were no significant differences in individual pairwise comparisons. 

We next sought to determine whether treatment disrupted sensorimotor gating in the P301L/COMTKO and htau models. Recent methodologies in evaluating PPI disruption with psychotomimetic agents have employed a strategy of differentiating high and low-baseline prepulse inhibitors and quantifying PPI disruption only in those mice that have relatively intact PPI at baseline (above the median), as mice with already diminished PPI integrity offer less of a target for disruption [73]. Employing this methodology, mice in these studies were divided by their baseline PPI. Those mice in each treatment group (scopolamine/reboxetine/saline) in which baseline PPI values were at or above the median for 12 dB PPI (an amplitude shown to be disrupted in the P301L/COMTKO model) [55] were included in the analyses, and the change in PPI in each mouse, from the baseline, within each treatment condition was assessed following 1 week of drug treatment or saline treatment. At the outcome, there were no significant differences in mean change in PPI in high-baseline inhibitors following treatment with scopolamine (*n* = 9), reboxetine (*n* = 8), or saline (*n* = 8) in P301L/COMTKO (F(2,22) = 2.01, *p* = 0.16) or htau mice treated with scopolamine (*n* = 8), reboxetine (*n* = 8), or saline (*n* = 8) (F(2,21) = 1.94, *p* = 0.17).

### 3.2. Neurobiology

#### 3.2.1. RNA-Seq 

Behavioral data from both tau models suggest the disruption of locomotor and startle response circuits in response to cholinergic antagonism that appeared to be more pronounced in P301L/COMTKO mice. For this reason, RNA-Seq was employed in the P301L/COMTKO model in order to evaluate differentially expressed genes and compare the cellular transcriptome in scopolamine versus saline-treated mice, looking, specifically, for changes in the expression of dopamine [74] or muscarinic [75] cholinergic pathway genes in P301L/COMTKO mice with relevance to these phenotypes. In the dopamine pathway, DRD3, the gene encoding the D3 receptor, was increased in scopolamine-treated mice relative to saline treated mice (LFC = 1.37, Wald test *p* = 0.047, Appendix A), but it did not survive tests for false discovery within all expressed genes. In scopolamine-treated mice, there was also an increase in the expression of DDC, encoding dopa decarboxylase—the enzyme that catalyzes the decarboxylation of L-3,4-dihydroxyphenylalanine (DOPA) to dopamine (LFC = 0.396, *p* = 0.015)—that did not survive tests for false discovery. There were no significant differences in the expression of DRD2, DRD3, DRD4, or DRD5. In the muscarinic cholinergic pathway, the expression of the gene encoding the M1 muscarinic receptor, CHRM1, was decreased in scopolamine-treated mice (LFC = −0.260, *p* = 0.00099), but it did not survive tests of false discovery. The expression of the gene encoding the M4 muscarinic receptor, CHRM4, was significantly decreased in scopolamine-treated mice, relative to saline (LFC = −0.376, *p* = 1.69 × 10^−5^, adjusted *p* = 0.0097), and the only gene to survive tests for false discovery within all expressed genes utilizing the Benjamini–Hochberg method [76]. There were no differences in expression of CHRM2, CHRM3, or CHRM5 or in any genes encoding the enzymatic production or degradation of acetylcholine. 

#### 3.2.2. Tau and Locomotion 

As the muscarinic receptor system has been shown to regulate extracellular concentrations of soluble tau [60,77], in order to determine whether scopolamine treatment increased soluble tau in the striatum, a region of dense expression of muscarinic receptors [78] soluble tau was measured in the P301L/COMTKO mice treated with scopolamine or saline. Soluble, rather than insoluble, tau in these mice was quantified, as we have found that 7 days (and even shorter) is a sufficient period to impact changes in soluble tau but insufficient time to impact aggregated tau in the P301L/COMTKO model [65]. Soluble tau (μg/mg protein) in the striatum was significantly increased in scopolamine-treated mice (*t* = 2.643, *p* = 0.013, Figure 3A). As our previous studies have suggested that, in tau mouse models, those with the highest degree of tau transgene expression have the largest increases in locomotive behavior [55], we looked to see whether total soluble tau in the striatum was associated with increased locomotion and whether scopolamine increased locomotion via an interaction with tau. There was a strong relationship between soluble tau and locomotive distance traveled in the cohort of 30 mice treated with either scopolamine or saline (R^2^ = 0.25, *p* = 0.005, Figure 3B). Multiple linear regression, followed by ANOVA (sum of squares), was performed in order to determine whether there were interactions between scopolamine treatment, continuous variables at outcome- soluble tau concentrations in the striatum, and distance traveled in the open field. There were interactions between scopolamine and increased soluble tau (F(1,26) = 11.93, *p* = 0.0019, Figure 3B), as well as scopolamine and distanced traveled (F(1,26) = 28.61, *p* < 0.0001, Figure 3B). There was also a significant interaction between scopolamine treatment and levels of soluble tau, together with distance traveled (three-way interaction), which suggests that the hyperlocomotion observed with scopolamine treatment is partially mediated by its impact on concentrations of soluble tau (F(1,26) = 11.82, *p* = 0.002, Figure 3B).

#### 3.2.3. MicroPET

As locomotion data suggests that scopolamine stimulates locomotive striatal circuitry and gene expression data suggests that there may be an increase in DDC transcription, which did not survive tests of false discovery, and thus, elevated dopamine production in the absence of any discernable change in D2R expression in scopolamine-treated mice, we sought to determine whether there was increased dopamine neurotransmission at D2R. We utilized [^11^C] raclopride MicroPET to estimate the competitive displacement of raclopride with endogenous dopamine. In order to evaluate the impact of scopolamine on dopaminergic neurotransmission, at the D2 in ventral and dorsal striatum, a discrete cohort (not from the behavioral experiments above) of female P301L/COMTKO mice, also 4-months-old, were treated with scopolamine (*n* = 8) or saline (*n* = 9) for 7 days, and they were imaged 1 h after the last injection with [^11^C] raclopride MicroPET. There were no significant differences in glucose metabolism in the striatum, as measured with [^18^F] FDG (Appendix A). Although previous studies of scopolamine in non-human primates suggest that muscarinic agonism and antagonism both increase glucose metabolism diffusely in the brain, reflecting increased neuronal activity, the measured cortical activity and striatum was not a region of interest [79]. As predicted in the current study, in an ROI analysis, [^11^C] raclopride binding was decreased in scopolamine-treated P301L/COMTKO mice, relative to saline mice, in caudate-putamen (dorsal striatum) and nucleus accumbens (ventral striatum) (Appendix A). In order to drill down into these differences, SPM analysis was conducted for voxel-wise comparisons, with conditions of ^11^C-raclopride in scopolamine < saline at *p* < 0.01, which showed that, compared to saline mice, D2/D3R occupancy was increased (decreased binding of ^11^C-raclopride) in scopolamine-treated mice in regions (Figure 4A, yellow voxels below, *t* = 2.457 and *p* < 0.01, Appendix A) that overlapped with the caudate-putamen (red) and nucleus accumbens (black). Further, these differences were significant within caudate-putamen and nucleus accumbens when a mask, obtained from SPM analysis with [^11^C] raclopride in scopolamine < saline at *p* < 0.05, was applied to these regions (Figure 4B). As our gene expression data suggests, there are no changes in D2R expression (as the data suggests, D3R also binds raclopride [80] and may be more rather than less abundant in scopolamine-treated mice). These reductions in binding in scopolamine-treated mice, in the context of increased DDC, support the contention that the impact of cholinergic antagonism on psychosis-like behaviors is driven by striatal increases in dopaminergic tone.

## 4. Discussion

In these psychotomimetic studies, treatment with the non-selective muscarinic antagonist scopolamine [81], rather than the noradrenergic reuptake inhibitor reboxetine [58], was associated with increases in locomotion and acoustic startle response in P301L/COMTKO and htau models. Differential gene expression and imaging studies in P301L/COMTKO mice treated with scopolamine suggest that these changes may be driven by changes in M1 and M4 muscarinic receptor expression and/or increases in striatal dopamine signaling. The muscarinic regulation of striatal dopamine release is well established [82]. The impact of cholinergic antagonism on striatal dopamine in these experiments is consistent with data from an older human study of a similar design, in which the administration of scopolamine was followed by changes in [^11^C] raclopride imaging that reflected decreased D2/D3 receptor availability from increases in endogenous striatal dopamine [83], and with the quantification of dopamine, following scopolamine treatment, in a non-human primate study [84]. Our results suggest that the disruption in behaviors observed may be related to increases in dopaminergic signaling in the dorsal and ventral striatum [78]. As studies of differential gene expression and endogenous dopamine were conducted in P301L/COMKO mice, it is not clear to what degree the COMT deletion augmented the availability of dopamine following scopolamine that could contribute to the behavioral changes and D2/D3R raclopride displacement. Behavioral changes in the COMT-competent htau model were, in the same direction following scopolamine, suggesting similar mechanisms. While it did not survive tests of false discovery, differential gene expression data suggest that DRD3 gene expression was increased in treated mice, perhaps in response to increased endogenous dopamine. Owing to this receptor increase, it is possible that the reduced binding of D2/D3 antagonist raclopride [80] in scopolamine-treated mice may underestimate striatal dopamine in these experiments. These results have direct relevance to the preclinical modeling of the psychotic symptoms of AD that have been associated with alterations in striatal dopamine physiology, as well as with an increase in DRD3 receptor density in AD patients with a history of psychosis [32,33].

In scopolamine-treated P301L/COMTKO and htau mice, exploratory velocity (distance traveled/time) in the open field was significantly increased; additionally, kinematic analysis, utilizing automated behavioral recognition software, revealed that hopping frequency [63]—a qualitative change in gait that could be related to speed (different from jumping, a repetitive stereotype lacking a discernable ambulatory goal and characterized by a vertical motion)—was also significantly increased in scopolamine-treated P301L/COMTKO and htau mice, suggesting an impact on central locomotor systems. Hyperlocomotion and increased dopaminergic neurotransmission in the striatum, in response to scopolamine, is likely due to secondary influences on M1 and M4 muscarinic receptors. M1 receptor mutant mice have been shown to have increased basal locomotion, disrupted PPI, and striatal dopamine release [85,86], while targeted M4 receptor deletion in mice has also been shown to increase locomotion, disrupt PPI, and increase striatal dopamine [75,87]. It follows that the receptor antagonism, at either or both receptors, could have a similar influence. Additionally, as we observed a significantly decreased CHR4 and non-significantly reduced CHR1 gene expression in scopolamine-treated mice, it may be that reductions in M1 and M4 receptor expression contributed to an increase in dopamine signaling and locomotion. Increases in dopamine in the caudate-putamen, suggested by [^11^C]-raclopride Micro-PET, which result from these receptor mechanisms, would be expected to drive changes in locomotion, as has been suggested by recent studies of the selective activation of dorsal-striatal-projecting dopamine neurons [88]. 

While muscarinic antagonism increased locomotion, we observed a reduction in locomotion in reboxetine-treated animals. Consistent with our findings, in a rat model of spontaneous hypertension [89], reboxetine has been reported to induce a reduction in locomotion [90]. The reduction in velocity from reboxetine treatment may reflect its focal impact on catecholamine neurotransmission in the frontal cortex. Reboxetine is known to be a potent and selective norepinephrine reuptake inhibitor; [91] however, previous studies have found that reboxetine increases dopamine in the frontal cortex of mice, which is an effect exacerbated by COMT deletion [92]. In our prior microdialysis experiments in P301L models, we reported on similar surges in frontal cortical dopamine following reboxetine injections in COMT-competent P301L mice, which were also augmented by COMT deletion [58]. This increase in extracellular dopamine, in response to reboxetine, has been reported to be isolated to the frontal cortex to the exclusion of the striatum [93,94]. An explanation for the reduction observed in locomotion following reboxetine treatment may come from the divergent roles of dopamine signaling in the frontal cortex and striatum in mediating motor behaviors that are reflected in the differential expression of receptor subtypes in these regions that exert opposing influences on movement. Mice with the deletion of the dopamine D1 receptor have been found to be hyperlocomotive [95,96,97], indicating a functional role in inhibiting locomotion, while mice with D2 receptor deletion have been found to be hypolocomotive [74]. While the role played by the D1-like receptors in determining locomotive activity is complex, and there seems to be a discrepancy between pharmacologic studies and gene knockout studies (locomotor stimulation has been demonstrated following D1 agonist exposure that is absent in D1 knockouts) [98], it may be that, as D1 receptors are moderately expressed throughout the cortex [99,100], focal surges in frontal dopamine from reboxetine uniquely impact the cortical D1 receptors and reduce locomotion. Alternatively, it may be that non-striatal D2 receptors, expressed at low levels in the cortex [101], also attenuate locomotion in response to stimulation. Support for the dichotomous impact of increases in cortical-versus-striatal dopamine on locomotion comes from the deficiency of cortical dopamine [102] in childhood hyperactivity that requires the use of psychostimulants in treatment [103] and from evidence that, in dopamine transporter (DAT) knockout mice, treatment with methylphenidate increased extracellular dopamine in the prefrontal cortex was associated with reductions in locomotion [104].

We found that scopolamine increased startle amplitude in P301L/COMTKO mice relative to reboxetine treatment. In rodents, compounds that are PPI disrupting tend to be acoustic startle response potentiating [105,106]. The acoustic startle reflex is a rapid response to a sudden stimulus that likely functions to protect the organism from an attack from a predator, and the hub of the network that innervates cervical and spinal motor neurons, mediating the skeletal muscle response, is the caudal pontine reticular nucleus (PnC). PnC-startled neurons [107] are believed to be under direct inhibitory control of the pedunculopontine tegmentum (PPT), a component of the ascending reticular activating system [108]. The modulation of the acoustic startle reflex via the PPT depends on cholinergic input, and studies in rodents suggest that cholinergic agonists reduce startle response, while muscarinic antagonists, including scopolamine, enhance startle amplitude [109,110]. Importantly, we did not find that scopolamine was PPI disrupting in either tau model. Muscarinic antagonists (including scopolamine at the dose used in these experiments) have been shown to impair PPI in wildtype rodents [73,111,112], and scopolamine has been studied, for this reason, in models of antipsychotic rescue [26]. Increases in striatal dopamine signaling in the ventral striatum/nucleus accumbens has been shown to be associated with impaired PPI in rodents [113], and [^11^C]-raclopride imaging data, following scopolamine treatment, suggest increased ventral striatal dopamine signaling. We previously reported on a strong correlation between aggregated tau and PPI disruption in rTg(P301L)4510 mice [54], as well as the resistance of tau models to PPI rescue [55]. It may be that, in the context of tau-driven deficits, modest perturbations in dopaminergic neurotransmission are insufficient to register sensorimotor gating effects. Future experiments in tau models that focus on tau reduction as a strategy for PPI rescue, in comparison with conventional pharmacologic approaches, may provide clarity.

The increase in total soluble tau observed in the current report, in response to scopolamine treatment, is consistent with evidence linking M1 and M3 muscarinic cholinergic receptors and tau pathology [60,77]. This is significant as aggregated fibrillary tau pathology, known to have a topographical distribution that changes predictably over the course of AD, leading to the characteristic decline in cognitive capacities [114], is now believed to spread inter-neuronally in a prion-like fashion, partially, via the excretion and reuptake of soluble tau [115]. As a potential mechanism, it has recently been shown in neuronal culture that the reuptake of tau by neurons is mediated, in part, by muscarinic receptors, and it can be reduced up to 80% by atropine and the M1-specific antagonist pirenzepine [77]. Although it not possible to determine whether changes were strictly extracellular, increases in total soluble tau that were associated with scopolamine treatment in the current report may have been the effect of direct reuptake inhibition via M1 antagonism, or they may have been secondary to decreased M1 expression and, thus, there was less receptor availability for reuptake. While tau levels did not have any association with sensorimotor gating integrity, elevations in total soluble tau in the striatum was associated with increased locomotion in the P301L/COMTKO; although this direct association with soluble tau has not previously been shown, the finding is consistent with previous reports that have found tau models to be hyperlocomotive relative to non-human-tau expressing wildtype comparators [116,117]. The association between cholinergic disruption and tau pathology may have implications for other tauopathies that affect motor systems, including Progressive Supranuclear Palsy (PSP), a neurodegenerative movement disorder characterized, neuropathologically, by aggregated tau pathology in the globus pallidus, a region within the striatum [118], and by the loss of cholinergic projection neurons in the acoustic startle-mediating PPT [119]. Recently, a rat model of PSP was developed, comprising the viral deposition of human tau in the cholinergic neurons of the PPT, resulting in the spread of tau pathology along neural connections and the disruption of acoustic startle and motor behavioral paradigms [120]. Given the observed increases in soluble tau in the current report, as well as the ubiquity of the use of anticholinergic medications, it may be important to test the impact of muscarinic antagonists, such as scopolamine, on tau spread in models of seeding-based tauopathy [121].

While the use of amphetamine [122] to induce diffuse increases in dopamine [88] has remained predominant in preclinical models of psychosis employing locomotion and sensorimotor gating [123,124,125], the use of muscarinic antagonism, as a modeling strategy in tau models, is supported only by the experiments in the current report focused on relevant locomotor (rather than sensorimotor gating) outcomes [12]. The results suggest that selective M1 and M4 muscarinic agonists could be investigated for their ability to rescue the scopolamine-induced hyperlocomotion and hopping observed in order to elucidate receptor-specific mechanisms. There is evidence for the clinical utility of muscarinic agonists in the treatment of psychosis in AD and schizophrenia that avoid the extra-pyramidal effects of a D2R blockade [126]. An early treatment trial in a small number of treatment refractory schizophrenic patients employing xanomeline, an M1/M4 muscarinic agonist and derivative of arecoline, reported evidence that the intervention is superior to placebo in treating psychotic symptoms [127]. Xanomeline may also have utility in the treatment of AD psychosis and the reduction in cognitive impairment; [128] currently, there is an active phase III clinical trial of a formulation of this compound in combination with trospium—a peripherally restricted muscarinic antagonist that offsets cholinergic side-effects—that is evaluating its ability to prevent psychosis relapse in AD [129]. It is clear that cholinergic pathways may be important avenues to explore in the development of new treatments for psychotic symptoms in AD while avoiding the known deleterious consequences of a direct D2R blockade [130], but the further development of preclinical models of behavior and pathology that emerge from cholinergic deficits may be important corollaries to these efforts. 

There are limitations to this study. While the MicroPET data supports the mechanistic interpretations of the behavioral data—that dopaminergic signaling in the striatum is responsible for the increased locomotion and sensorimotor gating perturbations in tau mouse models—we only had the capacity to perform the imaging experiments in one group of mice, and owing to the logistical complexities of these experiments, the imaging was not done on the same cohort of mice that underwent behavioral testing. Additionally, the mice did not have imaging at the baseline, so the changes in D2R occupancy in each mouse before and after treatment could not be directly assessed but only inferred from differences in outcome. Imaging mice before and after anticholinergic treatment, including in other strains of mice, is warranted in order to investigate this mechanism more closely and in order to determine whether the presence of the COMT deletion influences the outcomes. As these studies were focused on the comparison between the behavioral effects of muscarinic antagonism and dopaminergic stimulation in tau models of AD, rather than on a comparison of the impact of muscarinic antagonism in tau mice with non-tau mice, wildtype mice were not employed. Although comparable results to what was observed with htau mice would be expected from experiments with COMT-competent wildtype mice, future locomotion studies with wildtype mice, in the absence of human tau pathology, are required for comparison to determine the extent to which concentrations of soluble tau in the striatum found in these experiments mediate the impact of scopolamine on locomotor activity in the presence of tau. While reboxetine has relevance in tau models for the evaluation of surges in frontal dopamine [58], alternative interventions that increase dopamine in the striatum via reuptake inhibition, such as amphetamine or cocaine [131], may make suitable interventions for comparison with effects of scopolamine. 

## Figures and Tables

**Figure 1 biomedicines-11-02091-f001:**
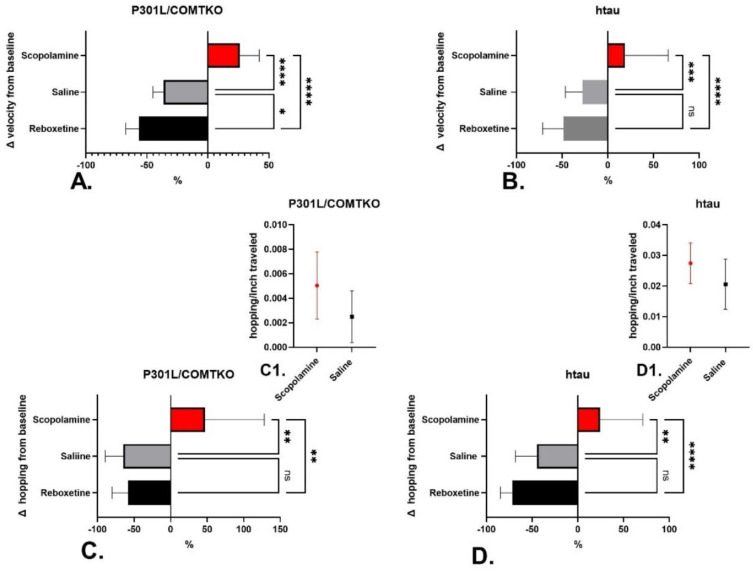
Percent changes in mean open field velocity (**A**,**B**) and hopping (**C**,**D**) (% change shown with 95% confidence interval) and differences in hopping frequency (**C1**,**D1**) (mean with 95% confidence interval shown) following 1 week of treatment with scopolamine (*n* = 15) or reboxetine (*n* = 15), in comparison with saline (*n* = 15), in P301L/COMTKO (**A**,**C**,**C1**) and htau (**B**,**D**,**D1**) mice. ANOVA followed by pairwise comparison * *p* < 0.05, ** *p* < 0.01, *** *p* < 0.001, **** *p* < 0.0001, ns = non-significant.

**Figure 2 biomedicines-11-02091-f002:**
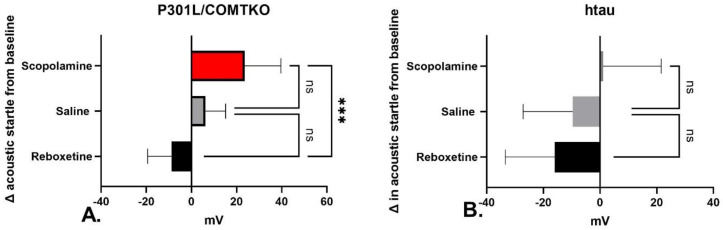
Mean change in the amplitude (mV, 95% confidence interval shown) of acoustic startle response to a 120 dB stimulus following 1 week of treatment with scopolamine (*n* = 15) or reboxetine (*n* = 15), in comparison with saline (*n* = 15), in (**A**) P301L/COMTKP and (**B**) htau mice. ANOVA followed by pairwise comparison *** *p* < 0.001, ns = non-significant.

**Figure 3 biomedicines-11-02091-f003:**
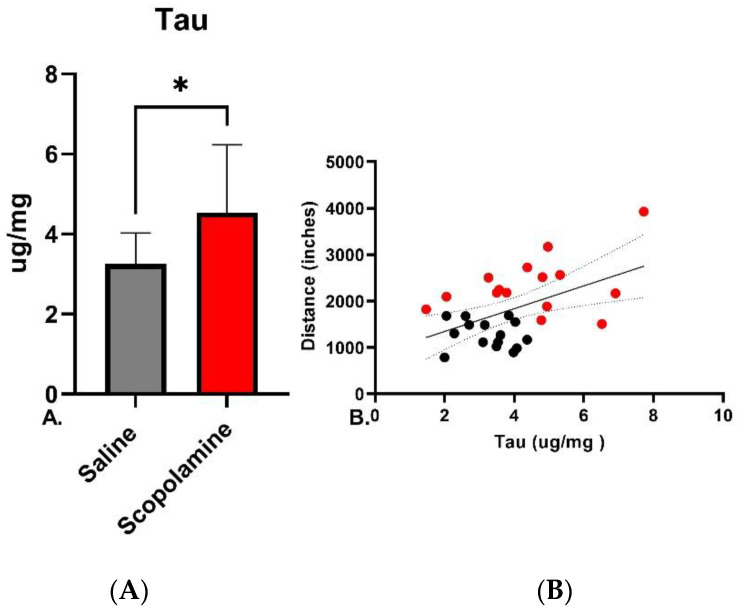
(**A**) Quantification of total soluble tau (DA31, μg/mg of protein) in the striatum of P301L/COMTKO mice treated for 1 week with scopolamine or saline determined via *t*-test, * *p* < 0.05 (**B**) Linear regression of distance traveled in the open field versus the total soluble tau in 30 mice (treated with either scopolamine or saline in behavioral experiments) with 95% CI shown, R^2^ = 0.25, *p* < 0.01. Multiple linear regression of scopolamine treatment status (red = scopolamine, black = saline) with the distance traveled and tau revealed significant interaction between scopolamine treatment, tau concentration, and distanced traveled *p* = 0.002.

**Figure 4 biomedicines-11-02091-f004:**
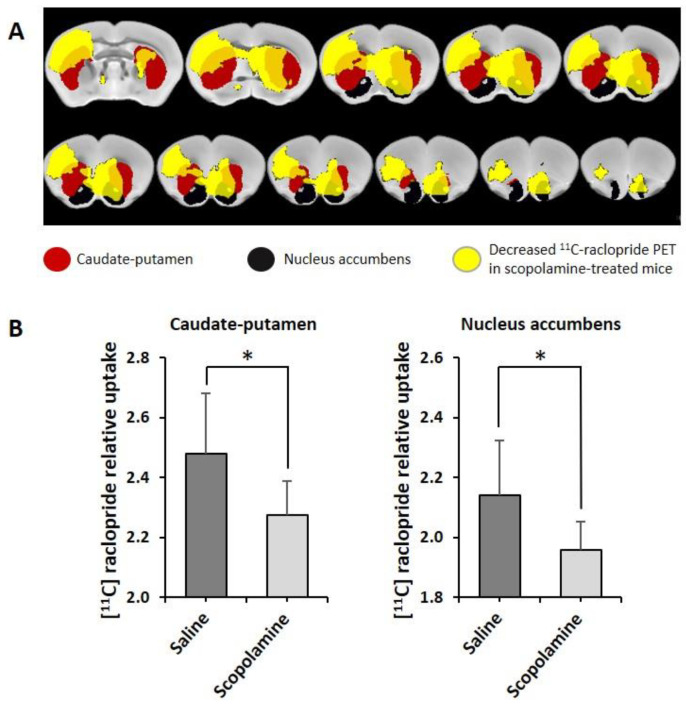
Differences in ^11^C-raclopride binding in scopolamine-treated P301L/COMTKO mice relative to saline-treated P301L/COMTKO mice. (**A**) Using the SPM voxel-wise analysis, a significant decrease in ^11^C-raclopride binding was detected in the scopolamine mice in the regions (yellow, *t* = 2.457 and *p* < 0.01) that overlapped with the caudate-putamen (red) and nucleus accumbens (black). Represented are two-dimensional displays of the decreased regions, caudate-putamen, and nucleus accumbens overlaid on a standard mouse MRI template (**B**). The ^11^C-raclopride relative uptakes were reduced in the scopolamine-treated mice, compared to saline mice (*p* < 0.05; Student *t* test), in the caudate-putamen (left) and nucleus accumbens (right) Mean values of ^11^C-raclopride PET in these regions were measured and normalized by the mean value of cerebellum in each image. Error bars represent standard deviations of the means; * *p* < 0.05.

## Data Availability

The data presented in this study are available on request from the corresponding author.

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
