# Peer review of "The Impact of Muscarinic Antagonism on Psychosis-Relevant Behaviors and Striatal [11C] Raclopride Binding in Tau Mouse Models of Alzheimer’s Disease"

_biomedicines, 2023, doi:10.3390/biomedicines11082091_

Round 1

Reviewer 1 Report

Interesting and well performed prychotomometic study of the locomotor effect of scopolaine and dopaminergic treatment of two mouse AD models (human-mutant P301L(COMTKO) and non-human mutant h-tau) with antimuscarinic scopolamine and DAergic reboxine. The first induced increased, the other decreased spontaneous locomotion. In P301/COMTKO/ model. scopolamine induced decreased M4 muscarinig receptor & increased DA D2 receptor expression, and increased extracellulR soluble tau in striatum (assessed by microPET & tau ELISA), prion-like transcellular spreading of which is mediated by muscarinic receptors. Techniques, image & assessment methods and results ar ereasonable. This obvious first experiment of scopolamin efficacy on AD-psychosis relevant behavior /locomotion in preclinical tau models appears of interest for further treatment of behavioral disorders in AD or other tauopathies.

Author Response

We thank the reviewer for this careful and detialed interrogation of the research aims, methodologies, and conclusions. 

Reviewer 2 Report

Re biomedicines-2491198

This is good study in experimental animals, dealing with the mechanisms of psychotic symptoms in AD. Such studies are important because they help improving our understanding on how anticholinergic agents may induce psychotic symptoms, why cholinergic agents may be helpful in treating such symptoms and, finally, they open new perspectives in the therapeutics of AD. In this context, the paper is welcome. The study is well conducted, experiments and statistics are excellent.

I have only one point, which I consider minor, but needs some discussion:

Page 2 line 92: Please explain briefly why the P301L mutant rodents offer a model of AD and not a model of FTDtau.

Author Response

We thank the reviewer for this excellent criticism, and pointing out the need for the clarification of the utility of mouse models of FTDtau in AD explicating AD biology. We have amended the manuscript with the following langauge and added the references below

"While the tauopathy of AD has been well characterized, as there are no known autosomal dominant mutations in the microtubule associated protein tau (MAPT) gene that are associated with familial AD, transgenic models that express mutant forms of human tau associated with frontotemporal dementia1 have been employed to study outcomes with relevance to the disease.2 "

  1. Hutton M, Lendon CL, Rizzu P, Baker M, Froelich S, Houlden H, Pickering-Brown S, Chakraverty S, Isaacs A, Grover A, Hackett J, Adamson J, Lincoln S, Dickson D, Davies P, Petersen RC, Stevens M, de Graaff E, Wauters E, van Baren J, Hillebrand M, Joosse M, Kwon JM, Nowotny P, Che LK, Norton J, Morris JC, Reed LA, Trojanowski J, Basun H, Lannfelt L, Neystat M, Fahn S, Dark F, Tannenberg T, Dodd PR, Hayward N, Kwok JB, Schofield PR, Andreadis A, Snowden J, Craufurd D, Neary D, Owen F, Oostra BA, Hardy J, Goate A, van Swieten J, Mann D, Lynch T, Heutink P. Association of missense and 5'-splice-site mutations in tau with the inherited dementia FTDP-17. Nature. 1998;393(6686):702-5. doi: 10.1038/31508. PubMed PMID: 9641683.
  2. Yokoyama M, Kobayashi H, Tatsumi L, Tomita T. Mouse Models of Alzheimer's Disease. Front Mol Neurosci. 2022;15:912995. Epub 20220621. doi: 10.3389/fnmol.2022.912995. PubMed PMID: 35799899; PMCID: PMC9254908.